# Associations between Vehicle Exhaust Particles and Ozone at Home Address and Birth Weight

**DOI:** 10.3390/ijerph17113836

**Published:** 2020-05-28

**Authors:** David Olsson, Christer Johansson, Bertil Forsberg

**Affiliations:** 1Department of Public Health and Clinical Medicine, Section of Sustainable Health, Umeå University, 901 87 Umeå, Sweden; bertil.forsberg@umu.se; 2Department of Environmental Science, Stockholm University, 106 91 Stockholm, Sweden; Christer.Johansson@aces.su.se; 3Environment and Health Administration, SLB-analys, 104 20 Stockholm, Sweden

**Keywords:** birth weight, traffic pollution, environmental epidemiology

## Abstract

We have studied the associations between exhaust particles and birth weight. Adjustments were made for ozone and potential confounding factors at the individual level. The study included all singletons conceived between August 2003 and February 2013 with mothers living in Greater Stockholm. We obtained record-based register data from the Swedish Medical Birth Register. Data concerning the parents were provided by Statistics Sweden. Exposure levels for nearly 187,000 pregnancies were calculated using a validated air quality dispersion model with input from a detailed emission database. A higher socioeconomic status was associated with higher levels of exhaust particles at the home address. In this region, with rather low air pollution levels, the associations between levels of exhaust particles and birth weight were negative for all three of the studied exposure windows (i.e., first and second trimester and full pregnancy). For the entire pregnancy, the linear decrease in birth weight was 7.5 grams (95% CI−12.0; −2.9) for an increase in exposure, corresponding to the inter quartile range (IQR = 209 ng/m^3^). We also found that the risk of being born small for gestational age increased with the level of exhaust particles in all three exposure windows, but these associations were not statistically significant.

## 1. Introduction

Restricted fetal growth, often studied as small for gestational age (SGA) or term low birth weight, is not only an important predictor of neonatal mortality and morbidity, but also a risk factor for adult mortality and morbidity. For mortality and morbidities related to cardiovascular disease, stroke, and type 2 diabetes, the risk increases continuously as birth weight decreases, even for those infants born within the normal (>2500 g) birth-weight range [1].

In addition to individual level maternal and social factors associated with restricted fetal growth, many studies have found adverse effects of air pollution on birth outcome. A recent systematic review, based on published cohort studies on prenatal exposure to fine particulate matter and adverse birth outcomes, included 43 studies from 2008 through 2017 [2]. This review found 23 studies investigating the impact of prenatal exposure to particulate matter, with an aerodynamic diameter of less than or equal to 2.5 µm (PM2.5), on the birth weights of infants, and 12 of them provided a significantly negative association between the level of exposure and birth weight. Among the nine included studies that investigated the impact of PM2.5 on SGA, five observed a significant association. Many of these studies on birth weight and SGA assessed the effects associated with exposure during different time windows, in particular, trimester-wise and for the full pregnancy. In these reviewed studies, overall the strongest effect of fine particles on birth weight and SGA was observed for exposure throughout the entire pregnancy [2]. An earlier review listed 10 studies of birth weight and PM2.5 exposure during the entire pregnancy, and 11 studies of SGA, but presented no combined estimates [3].

In another systematic review, the biggest pooled increase in risk for term low birth weight associated with 10 µg/m^3^ PM2.5 was observed for exposure during the entire pregnancy, however, the summary OR = 1.05 (95% CI 0.98; 1.12) was based on only four studies and was, therefore, not fully significant [4]. Seven studies presented estimates for IQR increments during the whole pregnancy and each trimester. The pooled OR estimate was above 1 only for the third trimester, OR = 1.03 (95% CI 0.98; 1.09), and the entire pregnancy, OR = 1.03 (95% CI 1.02; 1.03). Only a few birth weight studies have investigated associations with PM components, which is why these recent reviews do not include any such meta estimates [5].

The effects of ozone on fetal growth are much less studied than the effects of particle exposure. However, all six studies of ozone and birth weight included in a recent review reported statistically significant decrements; e.g., per 10 ppb over the entire pregnancy, the reduction ranged between −4.6 and −27.3 (n = 3) grams [3]. A later, very large study from China found a reduction in term birth weight of 5.93 grams for an IQR increase in ozone, corresponding to 12.15 µg/m^3^ [6].

The mixture of pollutants in ambient air is varying and complex, but vehicle exhaust particles and ozone typically show a negative correlation due to nitrogen oxide (NO) emissions reducing ozone levels when nitrogen dioxide (NO_2_) is formed [7]. This negative correlation suggests that the effects from local levels of exhaust particles on ozone should be studied in multi-pollutant models to avoid confounding effects. Ozone concentrations in northern Europe are also very dependent on season and weather.

The mechanisms by which fine particles can affect fetal growth are not well understood, but some hypotheses are quite well established. Reactive air pollutants can trigger systemic oxidative stress and inflammation, cause changes in the blood system and damage vascular endothelium, and thereby can decrease placental blood flow, disrupt transplacental oxygenation, cause placental oxidative stress and inflammation, and influence intrauterine growth restriction [8,9,10].

In previous studies from Stockholm, a capital region with low regional background levels of PM2.5 and minor industrial emissions, we first built only on temporal variation in air pollution concentrations, and found an association between first trimester ozone exposure and the incidence of preterm birth and pregnancy-induced hypertensive disorders [11,12]. In a later study of births in Stockholm in the period 1998 through 2007, we used the modeled level of nitrogen oxides (NOx) at the home address as a vehicle exhaust indicator and found the risk of pregnancy-induced hypertensive disorders to increase with exhaust exposure [13]. While most studies have used total PM2.5 estimated from monitoring stations, LUR models, and/or satellite observations, we wanted to study how birth weights would benefit from a reduction specifically in the mass concentration of exhaust particles, which only contributes to a small proportion of the total PM2.5 mass and may have a different concentration–response relation than PM2.5. Since local exhaust emissions mainly reduce ozone concentrations in this northern region, and we have observed the associations between ozone and birth outcomes, we chose to include two-pollutant model levels to investigate the associations between ambient air pollution at the residential address during pregnancy and birth weight (both actual birth weight and SGA).

## 2. Materials and Methods

### 2.1. Study Area and Population

We obtained record-based register data from the Swedish Medical Birth Register on all children born in Stockholm by mothers with an officially registered address in the Greater Stockholm area (including 26 municipalities), conceived between August 2003 and February 2013. Only singletons were included in the study.

### 2.2. Individual Characteristics

The information extracted from the Medical Birth Register includes information on the mother (e.g., age at delivery, parity, BMI, smoking habits, and health problems), collected by maternal care nurses during the pregnancy, the pregnancy and delivery (e.g., gestational length, type of labor and delivery mode, and complications), and the child (e.g., birth weight and size), reported from the delivery hospital. The national Medical Birth Register is nearly complete and the quality is good [14]. Register data concerning family situations, household incomes and both the mothers’ and fathers’ levels of education, together with geocoded home addresses at delivery were provided by Statistics Sweden.

### 2.3. Exposure Assessment

For all births we used the geocodes of the maternal home addresses to estimate the mean air pollution exposures during the first and second trimester, and over the entire pregnancy. The concentrations at the home address, due to local emissions of combustion particles and NOx, were calculated using a wind model and a Gaussian air quality dispersion model, both part of the Airviro Air Quality Management System (http://airviro.com). The dispersion calculations of road traffic emissions were performed on a fixed grid with a grid size of 100 × 100 m. Individual buildings and street canyons were not resolved but were treated using a roughness parameter. In an open area, the calculation height was 2 m above ground level. Over a city, the simulation reflected the concentrations at roof-top level height. The system, including the wind model, air quality dispersion models, emission database, and a database with monitoring data, is part of the Eastern Sweden Air Quality Management Association (www.slb.nu) and has been used in Stockholm for more than 20 years. It has provided exposure estimates for several epidemiological studies and health impact assessments [15,16,17].

The emission database, used as input data for the dispersion model calculations, is updated annually and contains detailed information about emissions from road and ferry traffic, petrol stations, industrial areas, and households [18]. The database is described in detail by Eneroth et al. (2015) [19]. The emissions from road traffic are described using emission factors (NOx and PM-exhaust) for different vehicles and road types, according to the European emission model HBEFA 3.1 (https://www.hbefa.net/e/index.html). For PM-exhaust, only road traffic contributions are included. Vehicle fleet emissions data are obtained from information on the distribution of vehicle types (e.g., diesel, petrol, and gas passenger cars and buses) and euro classes in the national vehicle registry. This provides the total mass of particles emitted from the exhausts of road traffic vehicles in the area but excludes secondary PM2.5 formed from exhaust emissions further away.

Ozone concentrations are indirectly calculated based on the modelling of NOx concentrations, as shown in Appendix A. It is assumed that the ozone concentration is linearly proportional to the measured difference in ozone concentration between the rural and urban background per unit of NOx at the central urban site. The basic chemistry behind this is that the ozone concentration in the city is controlled by transport from the surrounding areas into the city and by the removal of ozone due to its reaction with nitric oxide (NO).

Furthermore, the calculated NOx concentrations are scaled so that the calculated concentrations at the central site are set to the measured value. The dispersion model calculates hourly mean values that are scaled every hour based on the measurements. This means that the dispersion modelling of NOx is used to obtain the spatial contrasts in the ozone concentrations. The combined use of dispersion modelling and measured concentrations of ozone and NOx makes the amount of uncertainty in the estimated ozone concentrations very small.

### 2.4. Outcome Definition

We studied birth weight as a continuous variable, adjusted also for sex and gestational age in the model and small for gestational age (SGA) among singletons. SGA was defined as having a birth weight lower than the 10th percentile for the duration of gestation (in days), stratified by sex.

### 2.5. Variable Definitions

Potential confounders in the study of air pollution effects were maternal and paternal level of education, family income, family situation, parity, maternal smoking habits and body mass index (BMI) at the first antenatal visit, maternal region of origin, maternal age at delivery, and conception date. Maternal and paternal education were seven-level factor variables: pre-upper secondary school (<9 years), pre-upper secondary school (9 years), upper secondary school (<3 years), upper secondary school (3 years) post-upper secondary school (<3 years), post-secondary school (3 years), and postgraduate education. Family income was treated as a 10-level factor, based on deciles. Family situation was a three-level factor: cohabiting, single mother, and other. Parity was treated as a four-level factor: para one, para two, para three, and para four or higher. Maternal smoking habits at the first antenatal visit was a three-level factor: non-smoker, less than 10 cigarettes/day, and 10 or more cigarettes/day. Maternal BMI at first antenatal visit was a four-level factor, following the guidelines of the World Health Organization (i.e., underweight (BMI < 18.5), normal weight (18.5 < BMI < 25), overweight (25 < BMI < 30), and obese (BMI > 30). Maternal region of origin was treated as an 11-level factor: Northern Africa, Central and Southern Africa, Middle East, Southeast Asia, Oceania, North America, South and Central America, Western Europe, Eastern Europe, the Nordic Countries, and Sweden. Maternal age and conception date were treated as smooth functions. The annual proportion of households with a disposable income in the highest decile and the annual proportion of parents with a university education per municipality were used as linear predictors.

### 2.6. Statistical Analysis

Linear regression and log-binomial regression using the general additive model-function in the mgcv package in R was used to estimate the potential associations between the exposure variables and the outcomes of interest (birth weight and SGA, respectively). Unadjusted single and multi-pollutant models were fitted, as well as a model including all available potential confounding variables. In the analyses of birth weight, all models were adjusted for the duration of gestation. Initially potential non-linear associations were estimated with natural cubic splines, if the associations appeared to be linear; i.e., if the estimated degrees of freedom were less than 1.3, linear estimates were computed, and if the association appeared to be non-linear, estimates for a five-level factor, based on quintiles, were computed. To test statistical significance, 95% Confidence Intervals (95% CI) excluding the null were used.

Ethical approval was provided by the Regional Ethical Review Board at Umeå University (2014/393-31Ö).

## 3. Results

### 3.1. Temporal and Spatial Variations in Exposure

There was a significant reduction in the calculated exposure levels for both ozone and PM-exhaust during the period 2003–2013, as shown in Figure 1. Even though the share of the total number of diesel vehicles has increased during the period, from approximately 11% in 2003 to 31% in 2013 (passenger cars increased from 4.8% to 24%), the total road traffic emissions of NOx and PM-exhaust has reduced between 2003 and 2013 due to the renewal of the vehicle fleet. This is mainly due to lower emissions from new petrol vehicles, as shown by Krecl et al. (2017) [20]. The mean exposure to PM-exhaust is reduced by nearly half from the beginning to the end of the study period. The trend in calculated PM-exhaust is verified by measured decreasing trends in NOx concentrations and increasing trends in ozone at an urban background monitoring site in Stockholm [7].

Figure 2 shows the mean spatial distribution (all years) of modeled PM-exhaust and ozone concentrations in the model domain, Greater Stockholm. There is an inverse correlation between ozone and PM exhaust due to the NO titration of ozone, as mentioned before. The inverse correlation is also verified by measurements of NOx, the total particle number, and ozone at an urban background site in central Stockholm [7].

Full pregnancy exhaust particles levels ranged from 6.9 to 854 ng/m^3^ with an average of 202 ng/m^3^, and ozone ranged from 28.8 to 66.7 µg/m^3^ with an average of 54.6 µg/m^3^, as shown in Table 1.

### 3.2. Socioeconomic Indicators and Exposure

The highest levels of exhaust particles were observed near busy streets in the central parts of the city, where flats are often expensive. Mothers with low particle exposure most often live in suburbs at a distance from the city center. A high level of education and high family income was associated with higher levels of exhaust particles, as shown in Table 2. The mean exposure in the highest income decile was 74% higher than in the lowest decile. Mothers with a postgraduate degree had 69% higher mean exposure than those with less than nine years of school education. Ozone exposure was far less correlated to socioeconomic variables than exhaust particles.

### 3.3. Associations with Birth Weight

In total 186,912 births were included in the study. The average birth weight was 3531 g (standard deviation (SD) 541 g). The duration of gestation was, on average, 279 days (SD 12 days), and 4.4% of the births were preterm. The average maternal age was 31.4 years (SD 5 years), and 51.4% of the births were boys. Table 2 shows descriptive statistics for the births with complete data on the covariates included in the adjusted models. Children born to parents with a short education or with a mother who smoked in early pregnancy had a lower mean birth weight and were more likely to be SGA than other children.

Higher levels of first and second trimester and full pregnancy exhaust particles were associated with an increased risk of SGA in the unadjusted models without adjustments for potential confounders, as shown in Appendix B, but the associations were not statistically significant in the fully adjusted models, particularly after adjustment for parity, as shown in Figure 3. The adjusted prevalence rate ratio (PRR) was 1.01 (95% CI 0.99; 1.04) per interquartile range (IQR) increase in particle exposure during the first trimester (214 ng/m^3^), and 1.02 (95% CI 0.99; 1.05) for the entire pregnancy (209 ng/m^3^), as shown in Appendix B. There was a non-linear relationship between second trimester exhaust particles and SGA, with high rate ratios in the 4th (1.09 (95% CI 1.01; 1.17)) and 5th (1.07 (95% CI 0.99; 1.15)) quintiles, compared to the lowest quintile, as shown in Appendix B.

First trimester ozone was negatively associated with SGA both in the unadjusted model, as shown in Appendix B, and the full model, also adjusting for first trimester exhaust particles, as shown in Figure 3. In the linear model the PRR was 0.97 per IQR increase (19.4 µg/m^3^) and the 95% CI was (0.94; 1.00). There was no association between ozone exposure during the second trimester or entire pregnancy and SGA. The PRR for an IQR increase in exposure was 0.99 (95% CI 0.97; 1.02) for the second trimester (IQR = 20.2 µg/m^3^), and 1.01 (95% CI 0.98; 1.04) for the entire pregnancy (IQR = 8.0 µg/m^3^).

Higher levels of exhaust particles were associated with a non-linear decrease in birth weight in the unadjusted model, as shown in Appendix B, and a linear decrease in birth weight in the adjusted model, as shown in Figure 4, for the first trimester average levels, −6.5 g per IQR increase (95% CI −10.0; −3.0) in the adjusted model. Higher levels of exhaust particles during the second trimester were associated with a non-linear decrease in birth weight in the unadjusted model, but this was linear in the adjusted model, −8.7 g per IQR (213 ng/m^3^) increase (95% CI −12.3; −5.0). A higher level of exhaust particles during the entire pregnancy was associated with a non-linear decrease in birth weight in the unadjusted model, and a linear decrease in birth weight in the adjusted model, −7.5 g per IQR increase (95% CI −12.0; −2.9).

In the unadjusted models, higher levels of ozone were associated with an increased birth weight, regardless of the exposure window, as shown in Appendix B, but these associations did not remain in the adjusted models, as shown in Figure 4. The estimated effect of an IQR increase in ozone exposure during the first trimester was −1.4 g 95% CI (−5.2; 2.4). There appeared to be a non-linear association between second trimester ozone levels and birth weight. When comparing the highest quintile (>66.6 µg/m^3^) with the third quintile (>49.7 µg/m^3^ and ≤57.9 µg/m^3^), the difference was 11.2 g (95% CI 4.6; 17.8), and when comparing the lowest quintile (≤39.2 µg/m^3^) with the third quintile, the difference was 9.4 g (95% CI 2.6; 16.3), as shown in Appendix B. The estimated effect of an IQR increase in ozone exposure during the entire pregnancy was 3.7 g (95% CI −0.9; 8.2) in the adjusted model.

## 4. Discussion

In this large cohort study of air pollution and birth weight, vehicle exhaust emissions and the resulting exposure was studied with a much finer spatial resolution than is typical in similar studies. Streets and addresses in the same residential area as our study have different concentrations, reflecting the local situation. Many studies have used concentrations from monitoring stations that, for some study participants, were 25–30 kilometers away from their home, or modeled concentrations in grids that could be 12 × 12 km or larger, which meant that those associations were reflecting the effects of the regional and urban background concentrations.

The associations between levels of exhaust particles and birth weight were negative for all three exposure windows (first trimester, second trimester, and entire pregnancy). For the entire pregnancy, the linear decrease in birth weight was −7.5 g (95% CI −12.0; −2.9) per IQR increase (209 ng/m^3^). For the first trimester, the decrease was −6.5 g per IQR increase (95% CI −10.0; −3.0). For the second trimester, a 7.5 g (95% CI −12.0; −2.9) lower birth weight per IQR increase was observed. These results are comparable with the effect on birth weight per IQR increase in elemental carbon (200 ng/m^3^), found in Massachusetts, −9 g (95% CI −11; −8), adjusted for the remaining PM2.5 but not ozone [21]. An IQR effect of similar size, −7.8 g (95% CI −13.6; −2.0) was also observed for entire pregnancy PM2.5 (IQR = 1.9 µg/m^3^) in metropolitan Atlanta [22]. A European project pooled data on birth weight from eight mother–child cohorts conducted in areas where exposure to PM2.5 and the elemental composition of fine particles at the home address could be estimated from standardized but area specific land use regression (LUR) models [5]. Since the LUR models gave annual average concentrations, routine monitoring stations were used to temporally adjust the annual estimate to the time period of each pregnancy. The traffic related elements—copper, iron, and zinc—were not significantly associated with birth weight. In fact, copper and iron even had a positive regression coefficient. Sulphur was the only constituent that, besides total PM2.5 mass, was significantly associated with birth weight, and mainly reflected secondary particles, not vehicle exhaust. The reduction in birth weight was 32 g (95% CI 3; 29) per 10 µg/m^3^ PM2.5.

We also found SGA to increase with higher levels of exhaust particles in all three exposure windows, but these associations were not fully statistically significant. The adjusted PRR was 1.02 (95% CI 0.99; 1.05) for an IQR increase for the entire pregnancy exposure, which is in line with the presented combined estimate for term low birth weight from seven studies [4] where an IQR increment in PM2.5 over the entire pregnancy gave an OR = 1.03 (95% CI 1.02; 1.03).

A study from London found PM2.5 exhaust to adversely affect fetal growth when controlling for traffic-related noise [23]. For term SGA, an IQR increase in PM2.5 traffic exhaust (0.35 µg/m^3^) was associated with an OR = 1.02 (95% CI 1.00; 1.04).

Though less investigated, ozone is also reported to reduce fetal growth [3]. For ozone, we generally found smaller, inconsistent, and non-significant effects. For SGA, the PRR for an IQR increase in exposure for the entire pregnancy was 1.01 (95% CI 0.98; 1.04), as opposed to the 3.7 g (95% CI −0.9; 8.2) change in birth weight estimated for an IQR increase in ozone exposure during the entire pregnancy. Because there is very little ozone formation in this region, levels of ozone and vehicle exhaust show an inverse correlation in space and time due to a reduction in ozone when NO is oxidized. Higher ozone exposure generally means less exhaust exposure, which suggests that the two-pollutant models should be used to reduce the risk of confounding.

Adverse effects on birth weight and SGA were, as expected, observed for smoking during pregnancy and low socioeconomic status, such as short education, low income, and single mothers. Most socioeconomic factors were negatively correlated with the level of exhaust particles at the home address, which means that protective socioeconomic conditions are associated with higher exposure to PM-exhaust during pregnancy. Without adjustment for socioeconomic factors, these confounders tend to mask the adverse effects of vehicle exhaust.

One strength of our study design is that we used state register data, allowing us to include all mothers and births as opposed to birth cohort studies, which include a selection process where the included samples may not fully represent the study population. Moreover, this method excludes potential reporting bias, since no data on risk factors during the pregnancy were reported after the child was born.

Another strength of our study is the exposure assessment, using a high spatial resolution, with the combination of geocoded home addresses, fine-scale emission data, and the dispersion calculations of vehicle exhaust emissions, performed with a grid size of 100 × 100 m. One limitation is, of course, that other traffic related pollutants not included, as CO, NO_2_, and road dust, are highly correlated with PM-exhaust at this scale. This means that these components may partly have contributed to the observed associations between PM-exhaust and birth weight. Another limitation is that we were not able to take into account the changes in residential addresses during pregnancy. This would lead to exposure misclassification, which may lead to bias in both directions; however, it has been shown, in a Swedish population, that the study population tended to move to areas with similar levels of air pollution [24]. However, since we did not send any questionnaires to parents to collect data, we do not know anything about the indoor environment, such as if the father smoked indoors during the pregnancy, which would be a risk factor for term low birth weight in a recent study [25].

We also understand that, in our residential areas with higher average ozone levels, there is more greenness, which we did not study, but which has been associated with higher birthweight and lower odds and SGA [26].

## 5. Conclusions

We found local levels of vehicle exhaust particles at the residential address during pregnancy to be associated with a lower birth weight in Greater Stockholm, Sweden, an area with relatively low levels of air pollution. We were able to adjust for many potential confounding factors at the individual level, but most socioeconomic risk indicators were negatively correlated with the level of exhaust particles at the home address, since residences in the city center are expensive. The associations between the levels of exhaust particles and birth weight were negative for all three studied exposure windows. We also found the risk of being born small for gestational age to increase with the level of exhaust particles, but these associations were not fully statistically significant.

## Figures and Tables

**Figure 1 ijerph-17-03836-f001:**
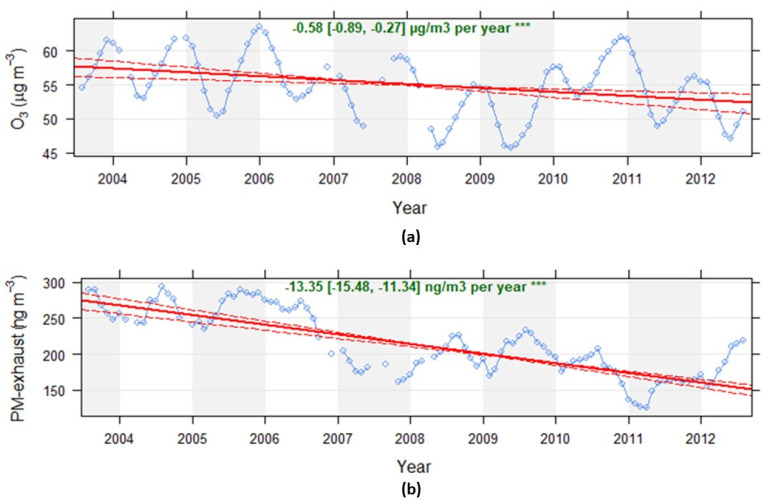
Calculated monthly mean exposure concentrations of ozone (**a**) and exhaust particles (**b**) for all mothers and all trimesters included in the study. The trends are based on monthly mean values with at least 75% data capture. The Openair package “TheilSen” function in R accounting for autocorrelations and bootstrap simulations have been used to obtain the trends (*** means p < 0.001).

**Figure 2 ijerph-17-03836-f002:**
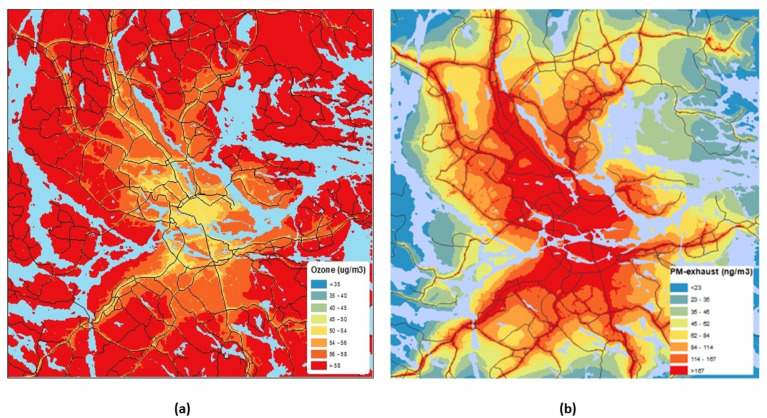
Spatial distribution of ozone (**a**) and exhaust particles (**b**) concentrations in the model domain encompassing all mothers. The concentrations were calculated at a resolution of 100 × 100 m and the figures show the averages of all hours during the whole period, 2003–2013.

**Figure 3 ijerph-17-03836-f003:**
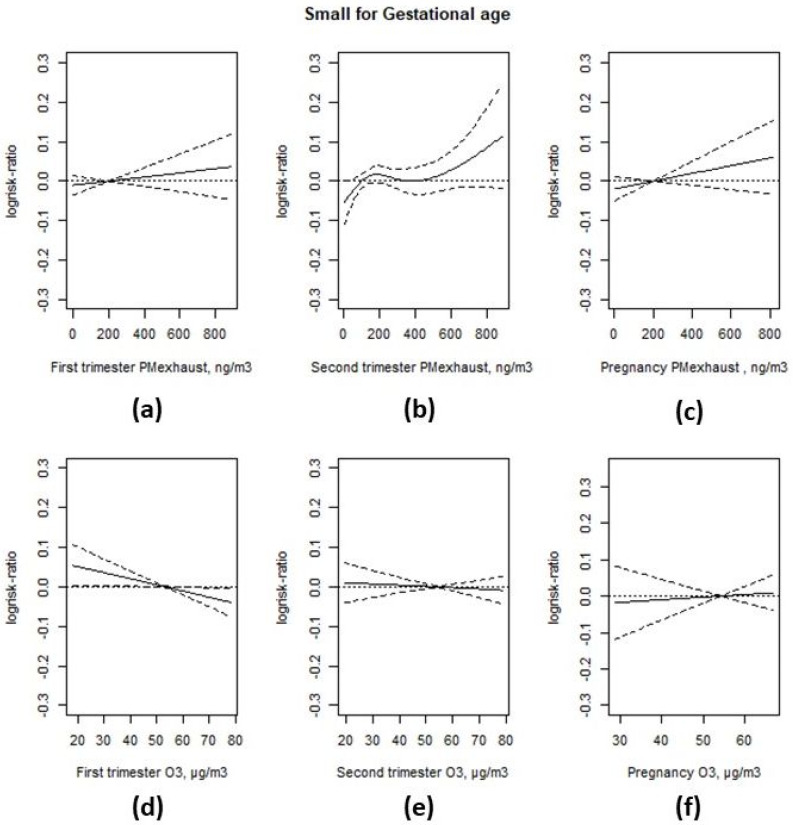
Associations between PM-exhaust (**a**–**c**), ozone (O_3_) (**d**–**f**) during different stages of gestation and SGA from two-pollutant models. All models were adjusted for maternal and paternal level of education, family income, area level income and education, family situation, parity, maternal smoking habits, body mass index (BMI) at first antenatal visit, maternal region of origin, maternal age at delivery, and conception date.

**Figure 4 ijerph-17-03836-f004:**
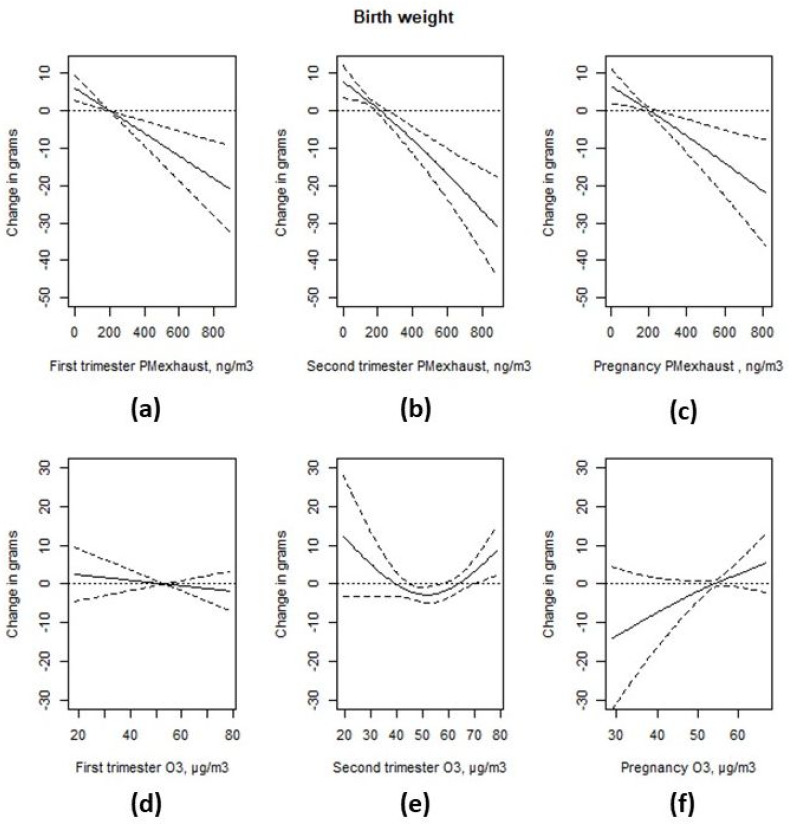
Associations between PM-exhaust (**a**–**c**), ozone (O_3_) (**d**–**f**) during different stages of gestation and birth weight from two-pollutant models. All models were adjusted for maternal and paternal level of education, family income, area level income and education, family situation, parity, maternal smoking habits, body mass index (BMI) at first antenatal visit, maternal region of origin, maternal age at delivery, duration of gestation, and conception date.

**Table 1 ijerph-17-03836-t001:** Summary table of the distribution of air pollution exposure.

	Min	Median	Max	Interquartile Range
First trimester PM-exhaust (ng/m^3^)	3	159	886	214
Second trimester PM-exhaust (ng/m^3^)	3	159	882	213
Pregnancy PM-exhaust (ng/m^3^)	7	160	854	209
First trimester ozone (µg/m^3^)	18.5	52.8	78.4	19.4
Second trimester ozone (µg/m^3^)	19.3	53.6	78.6	20.2
Pregnancy ozone (µg/m^3^)	28.8	54.9	66.7	8.0

**Table 2 ijerph-17-03836-t002:** Summary table of outcomes and exposure levels during full pregnancy in different sub-groups of the study population.

	Factor Level	Subjects (%)	Small for Gestational age, %	Birth Weight, g (SD)	PM-Exhaust, ng/m^3^ (SD)	Ozone, µg/m^3^ (SD)
Maternal level of education	pre-upper secondary school (<9 years)	4021 (2.7)	13.5 (34.2)	3444.2 (528.1)	145.8 (99.1)	55.6 (5.0)
	pre-upper secondary school (9 years)	9981 (6.7)	11.4 (31.8)	3481.0 (549.5)	150.9 (129.4)	55.7 (5.3)
	upper secondary school (<3 years)	13,571 (9.2)	10.5 (30.7)	3510.2 (550.0)	165.2 (139.5)	55.8 (5.5)
	upper secondary school (3 years)	34,643 (23.4)	9.2 (28.9)	3547.0 (532.3)	170.3 (149.5)	55.2 (5.7)
	post-upper secondary school (<3 years)	20,312 (13.7)	9.7 (29.6)	3544.8 (534.5)	206.6 (160.6)	54.3 (5.8)
	post-secondary school (3 years)	63,605 (42.9)	9.0 (28.6)	3563.0 (522.8)	233.7 (165.1)	53.4 (5.9)
	postgraduate education	2103 (1.4)	9.4 (29.2)	3552.8 (524.4)	244.7 (166.4)	53.1 (6.0)
Paternal level of education	pre-upper secondary school (<9 years)	3589 (2.4)	14.7 (35.4)	3426.3 (530.9)	137.9 (92.3)	55.7 (4.9)
	pre-upper secondary school (9 years)	13,365 (9.0)	10.8 (31)	3507.6 (538)	151.8 (132.2)	55.6 (5.4)
	upper secondary school (<3 years)	21,049 (14.2)	10.2 (30.2)	3530.5 (551)	163.5 (139.4)	55.7 (5.5)
	upper secondary school (3 years)	33,891 (22.9)	9.1 (28.7)	3545.0 (532.1)	176.7 (151.6)	54.9 (5.7)
	post-upper secondary school (<3 years)	23,358 (15.8)	9.6 (29.5)	3550.0 (531.4)	212.0 (161.5)	54.2 (5.8)
	post-secondary school (3 years)	50,052 (33.8)	9.0 (28.6)	3560.7 (520.5)	243.2 (166.2)	53.2 (6.0)
	postgraduate education	2932 (2.0)	8.9 (28.5)	3559.1 (532.8)	249.2 (167.1)	53.1 (6.0)
Family income, deciles	1st	12,135 (8.2)	14.0 (34.7)	3434.5 (532.8)	154.3 (120.4)	54.8 (5.3)
	2nd	14,744 (9.9)	12.1 (32.6)	3480.7 (536.2)	162.9 (128.7)	54.8 (5.4)
	3rd	16,036 (10.8)	10.6 (30.7)	3518.9 (534.2)	169.3 (136.9)	54.5 (5.5)
	4th	16,979 (11.5)	8.9 (28.4)	3550.2 (533.1)	175.8 (145.7)	54.5 (5.7)
	5th	16,646 (11.2)	8.9 (28.5)	3554.5 (539.7)	186.5 (152.0)	54.6 (5.8)
	6th	15,889 (10.7)	8.9 (28.5)	3568.7 (532.8)	205.8 (161.4)	54.4 (6.0)
	7th	15,212 (10.3)	8.5 (27.9)	3574.0 (526.5)	219.5 (164.3)	54.2 (6.1)
	8th	14,675 (9.9)	8.0 (27.1)	3579.0 (531.6)	234.8 (169.6)	54.1 (6.2)
	9th	13,219 (8.9)	7.8 (26.8)	3581.0 (514.5)	248.1 (174.4)	54.0 (6.1)
	10th	12,701 (8.6)	8.7 (28.1)	3574.9 (513.3)	267.5 (182.1)	53.7 (6.2)
Maternal smoking	Non-smoker	141,859 (95.7)	9.3 (29)	3549.6 (530.0)	203.8 (159.2)	54.3 (5.8)
	<10 cigarettes/day	4977 (3.4)	15.6 (36.3)	3415.1 (555.2)	144.9 (128.0)	56.0 (5.3)
	≥10 cigarettes/day	1400 (0.9)	16.5 (37.1)	3336.7 (542.9)	131.2 (118.3)	56.3 (5.3)
Family situation	Co-habiting with father	142,456 (96.1)	9.4 (29.2)	3546.1 (531.6)	201.7 (159.0)	54.4 (5.8)
	Living alone	1575 (1.1)	13.5 (34.2)	3459.7 (541.6)	180.4 (134.1)	54.6 (5.4)
	Other	4205 (2.8)	13.7 (34.4)	3471.8 (532.6)	189.9 (147.8)	54.7 (5.6)
Maternal BMI	Underweight	4138 (2.8)	17.7 (38.2)	3321.2 (491.7)	217.1 (166.0)	53.9 (5.9)
	Normal Weight	100,208 (67.6)	10.0 (30.0)	3518.8 (513.6)	215.7 (164.0)	54.0 (5.9)
	Overweight	31,681 (21.4)	7.8 (26.8)	3610.9 (549.5)	173.2 (142.4)	55.1 (5.6)
	Obese	12,209 (8.2)	8.2 (27.5)	3641.1 (601.2)	148.5 (124.7)	55.6 (5.4)
Maternal region of origin	Central and Southern Africa	4556 (3.1)	15.8 (36.5)	3447.7 (564.9)	169.8 (102.4)	55.0 (5.1)
	Oceania	100 (0.1)	11.0 (31.4)	3583.3 (538.2)	236.3 (171.1)	53.8 (5.9)
	Eastern Europe	8404 (5.7)	11.1 (31.4)	3509.1 (526.7)	161.6 (125.3)	55.3 (5.4)
	Middle East	9841 (6.6)	14.6 (35.3)	3398.3 (507.1)	147.4 (111.5)	55.7 (5.1)
	Northern Africa	1426 (1.0)	11.2 (31.6)	3496.3 (522.0)	151.9 (106.0)	55.4 (5.0)
	North America	624 (0.4)	9.1 (28.9)	3552.3 (495.5)	247.7 (170.4)	53.5 (6.3)
	Other Nordic countries	2623 (1.8)	8.8 (28.4)	3567.5 (534.8)	201.4 (163.0)	54.7 (6.0)
	South and Central America	3202 (2.2)	9.8 (29.7)	3485.2 (551.6)	190.9 (143.4)	54.6 (5.6)
	Southeast Asia	5317 (3.6)	16.2 (36.9)	3349.6 (536.5)	194.9 (144.2)	54.4 (5.7)
	Sweden	110,176 (74.3)	8.4 (27.7)	3574.0 (527.8)	210.7 (165.5)	54.1 (5.9)
	Other Western European	1967 (1.3)	11.7 (32.2)	3513.6 (517.6)	225.0 (169.0)	53.8 (6.1)

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
