# Peer review of "Associations between Vehicle Exhaust Particles and Ozone at Home Address and Birth Weight"

_ijerph, 2020, doi:10.3390/ijerph17113836_

Round 1
Reviewer 1 Report
This is a large, registry-based study of infant birth weight and maternal exposure to vehicle exhaust-related particles during pregnancy. Important covariates are appropriately adjusted and possible non-linear associations are evaluated. The relationship between particulate air pollution during pregnancy and infant birth weight has been studied rather extensively in other settings. Therefore, the Introduction should explain more clearly why additional study of this topic is needed, and how their exposure model improves upon previous work (including co-pollutant ozone).
Major comments
- While the exposure model has been successfully applied in other epidemiologic settings, the quantities estimated here require clarification. The abstract provides a change in birth weight for each IQR increase in “exhaust particles”, and gives the IQR as 209 ng/m3. However, it is unclear exactly which particles these units refer to. Is this quantity comparable to PM reported in other studies? Or a subset of specifically vehicle-derived PM? Additionally, please provide a table describing the distribution of particle concentrations in the study population in each period/trimester of pregnancy (median and IQR in addition to minimum, maximum) for comparison with other studies. Table 1 currently only provides exposure information stratified by covariates, and lists “Exhaust PM10”, which is not a term used anywhere else in the manuscript. The title of the manuscript should be clarified as well to better describe exposures and include ozone.
- If exposure is assigned to the mother’s address only at the time of delivery, please describe the potential for exposure misclassification due to changes in residential address during pregnancy.
- Methods, lines 96-101: Please explain if these important outcome data and covariates are recorded by the mother or by a medical provider. Is there any published data on the accuracy of such information in the Swedish Medical Birth Register?
- Results, Table 1: Please include both the number and percentage of participants in each category. Are the exposure levels provided for the full pregnancy only? Please also describe the distribution of infant sex, maternal age, parity, and infant gestational age at birth, and the number and percentage of preterm births included in the study.
- Figure 3 and Figure 4 caption should state that models were additionally adjusted for the other pollutant (Ozone or PM-exhaust), if this is the case.
- Results: Please provide all regression results in a table (not only figures or text), with footnotes stating clearly all variables adjusted in the model.
- Why were models adjusted for conception date? Does this possibly over-adjust for seasonal effects by removing all temporal variation in exposure?
- Results, lines 251-253: If there is a non-linear (U-shaped) association between 2nd trimester ozone and birth weight, then comparing the highest to lowest decile does not seem to be the most appropriate method for evaluating this. For example, does the middle tertile differ from the highest/lowest?
- Discussion, lines 288-289: Unclear what study this last sentence refers to. Is traffic noise data available in the present study?
- Discussion: As the co-adjustment for ozone is a substantial contribution of this study, please discuss the interpretation and importance of these results in the Discussion.
- Conclusions, lines 306-307: It is not appropriate to use causal language with these observational data – please remove the phrase “reduce birth weight”.
- Conclusions: This is the first mention of the “relatively low levels of air pollution” in Stockholm – this should be described in the Introduction to justify the study.
Minor comments
- Abstract, lines 17-18: It is unclear what is meant by “socioeconomic risk indicators”. While this is clarified later in the manuscript, I suggest stating clearly in the abstract that higher socioeconomic status was associated with higher exposure (perhaps contrary to expectation).
- Introduction, lines 52-58: This paragraph does not clearly contribute to the justification for the study and should be moved to the Discussion or omitted.
- Introduction, line 76-79: Similarly, this paragraph is confusing because thyroid hormones are not part of the current study.
- Results, lines 188-189: Suggest using the term “inverse correlation” rather than “anti-correlation”.
- Results, line 203: “10th income decile” is not clear – replace with “highest income decile.”
- Discussion, lines 257-262: While this information is important, it is not the main finding of the study. I suggest moving it to later in the Discussion.
- Discussion, lines 295-297: There are limitations to using registry-based data as well, which are not collected for research purposes. For example, maternal smoking is likely to be under-reported. Do you have any information on data quality in this registry?
Author Response
This is a large, registry-based study of infant birth weight and maternal exposure to vehicle exhaust-related particles during pregnancy. Important covariates are appropriately adjusted and possible non-linear associations are evaluated. The relationship between particulate air pollution during pregnancy and infant birth weight has been studied rather extensively in other settings. Therefore, the Introduction should explain more clearly why additional study of this topic is needed, and how their exposure model improves upon previous work (including co-pollutant ozone).
ANSWER: We have now commented and motivated our study in end of the Introduction. There are many studies of PM2.5 and birth weight, in their recent review Yoan et al (2019) identified such 23 studies. However, we study the association with the concentrations of locally emitted exhaust particles using dispersion modeling with high spatial resolution. Other studies have used total PM2.5 estimated from monitoring stations, LUR models and/or satellite observations. In those studies the mean and maximum exposure of cause becomes much higher. Our study is more relevant when the question is, how would birth weights benefit from a quantified reduction in exhaust emissions.
Major comments
- While the exposure model has been successfully applied in other epidemiologic settings, the quantities estimated here require clarification. The abstract provides a change in birth weight for each IQR increase in “exhaust particles”, and gives the IQR as 209 ng/m3. However, it is unclear exactly which particles these units refer to. Is this quantity comparable to PM reported in other studies? Or a subset of specifically vehicle-derived PM?
ANSWER: We hoped that this was explained under Exposure (Section 2.3), but have added one sentence to clarify.
Additionally, please provide a table describing the distribution of particle concentrations in the study population in each period/trimester of pregnancy (median and IQR in addition to minimum, maximum) for comparison with other studies. Table 1 currently only provides exposure information stratified by covariates, and lists “Exhaust PM10”, which is not a term used anywhere else in the manuscript.
ANSWER: The air pollution concentrations in the study population are now included in the new Table 1, and exhaust pm has been relabeled in what is now Table 2.
The title of the manuscript should be clarified as well to better describe exposures and include ozone.
ANSWER: Ozone has been added in the title.
- If exposure is assigned to the mother’s address only at the time of delivery, please describe the potential for exposure misclassification due to changes in residential address during pregnancy.
ANSWER: We now comment the potential limitation that we do not have more than one address in the Discussion.
- Methods, lines 96-101: Please explain if these important outcome data and covariates are recorded by the mother or by a medical provider. Is there any published data on the accuracy of such information in the Swedish Medical Birth Register?
ANSWER: Details on the information sources and a reference have been added in Section 2.2.
- Results, Table 1: Please include both the number and percentage of participants in each category. Are the exposure levels provided for the full pregnancy only? Please also describe the distribution of infant sex, maternal age, parity, and infant gestational age at birth, and the number and percentage of preterm births included in the study.
ANSWER: We have added this information in the new Table 2, and a few sentences describing background variables has been added on lines 215-218.
- Figure 3 and Figure 4 caption should state that models were additionally adjusted for the other pollutant (Ozone or PM-exhaust), if this is the case.
ANSWER: Yes, this has been added.
- Results: Please provide all regression results in a table (not only figures or text), with footnotes stating clearly all variables adjusted in the model.
ANSWER: A new table has been added in Appendix B.
- Why were models adjusted for conception date? Does this possibly over-adjust for seasonal effects by removing all temporal variation in exposure?
ANSWER: We model for time trends and seasonality this way, but the variable is not allowed to be very fluctuating. Seasonality may be related to seasonal variation in eating habits, temperature, viral infections and allergies, and is better modeled using conception date rather than birth date.
- Results, lines 251-253: If there is a non-linear (U-shaped) association between 2nd trimester ozone and birth weight, then comparing the highest to lowest decile does not seem to be the most appropriate method for evaluating this. For example, does the middle tertile differ from the highest/lowest?
ANSWER: We have rerun the models with non-linear associations now using five-level factors, based on quintiles. This is now mentioned under Methods and the new results have been included. Tables have been added in appendix B, reporting the regression coefficients for all factor levels.
- Discussion, lines 288-289: Unclear what study this last sentence refers to. Is traffic noise data available in the present study?
ANSWER: We don’t have noise data, and have dropped the reference.
- Discussion: As the co-adjustment for ozone is a substantial contribution of this study, please discuss the interpretation and importance of these results in the Discussion.
ANSWER: This has been included in the Discussion.
- Conclusions, lines 306-307: It is not appropriate to use causal language with these observational data – please remove the phrase “reduce birth weight”.
ANSWER: We have changed and now say “… to be associated with a lower birth weight.”
- Conclusions: This is the first mention of the “relatively low levels of air pollution” in Stockholm – this should be described in the Introduction to justify the study. ANSWER: Yes, we now mention this both in the Abstract and in the Introduction.
Minor comments
- Abstract, lines 17-18: It is unclear what is meant by “socioeconomic risk indicators”. While this is clarified later in the manuscript, I suggest stating clearly in the abstract that higher socioeconomic status was associated with higher exposure (perhaps contrary to expectation).
ANSWER: Yes, we have changed accordingly.
- Introduction, lines 52-58: This paragraph does not clearly contribute to the justification for the study and should be moved to the Discussion or omitted.
ANSWER: Yes, we have included most of it in the Discussion.
- Introduction, line 76-79: Similarly, this paragraph is confusing because thyroid hormones are not part of the current study.
ANSWER: Yes, we excluded it.
- Results, lines 188-189: Suggest using the term “inverse correlation” rather than “anti-correlation”.
ANSWER: Yes, we have changed accordingly.
- Results, line 203: “10th income decile” is not clear – replace with “highest income decile.”
ANSWER: Yes, we have changed accordingly.
- Discussion, lines 257-262: While this information is important, it is not the main finding of the study. I suggest moving it to later in the Discussion.
ANSWER: Yes, we have changed accordingly.
- Discussion, lines 295-297: There are limitations to using registry-based data as well, which are not collected for research purposes. For example, maternal smoking is likely to be under-reported. Do you have any information on data quality in this registry?
ANSWER: Most of the registry data we use are objective data (e.g. BMI, education, income), but of course smoking can be under-reported as in all kinds of interviews. However, this information is prospectively collected, and cannot be biased because of the birth outcome.
Reviewer 2 Report
This work is interesting and the quality is fine. Comments are as follows:
1. TITLE. "Associations between vehicle exhaust particles at home address and birth weight" may be not consistent with the contents of the work, because O3 was also investigated. How about "vehicle exhaust pollution" or "vehicle exhaust pollutants"?
2. DISCUSSION.
The authors found that "Most socioeconomic risk indicators were negatively correlated with level of exhaust particles at home address, which means that protective socioeconomic conditions are associated with higher exposure to PM-exhaust during pregnancy". This is interesting! However, socioeconomic risk indicators or low Socio-Economic Status (SES) may be related to exposure to high levels of other types of air pollution, such as particulate air pollution or industry-related air pollution. Please refer to "Deng et al. Parental stress and air pollution increase childhood asthma in China. Environmental Research 2018, 165: 23-31".
How about LIMITATIONS? The authors should mention some limitations. My main concern about this work is that they did not consider other type of air pollution or indoor air pollution, for example "Lu et al. Combined effects of ambient air pollution and home environmental factors on low birth weight. Chemosphere 2020, 240: 124836".
Author Response
- TITLE. "Associations between vehicle exhaust particles at home address and birth weight" may be not consistent with the contents of the work, because O3 was also investigated. How about "vehicle exhaust pollution" or "vehicle exhaust pollutants"?
ANSWER: We have added ozone in the title.
- DISCUSSION.
The authors found that "Most socioeconomic risk indicators were negatively correlated with level of exhaust particles at home address, which means that protective socioeconomic conditions are associated with higher exposure to PM-exhaust during pregnancy". This is interesting! However, socioeconomic risk indicators or low Socio-Economic Status (SES) may be related to exposure to high levels of other types of air pollution, such as particulate air pollution or industry-related air pollution. Please refer to "Deng et al. Parental stress and air pollution increase childhood asthma in China. Environmental Research 2018, 165: 23-31".
ANSWER: This is not a problem. The study area, the Swedish capital region, has except traffic very low emissions. There are no metal, pulp or petrochemical industries. The areas with lower socioeconomic status are typically areas with rental housing (often municipally owned), district heating, electric stoves and no use of solid fuels.
How about LIMITATIONS? The authors should mention some limitations. My main concern about this work is that they did not consider other type of air pollution or indoor air pollution, for example "Lu et al. Combined effects of ambient air pollution and home environmental factors on low birth weight. Chemosphere 2020, 240: 124836".
ANSWER: We have in the Discussion added comments about study limitations.
We commented lack of other important outdoor sources above. We don’t have information on exposure to ETS, but expect maternal smoking, parental education and other SES variables to partly adjust for passive smoke at home during the pregnancy. Higher vehicle exhaust exposure is correlated with higher wealth and educational level, why confounding from a poor home environment unlikely could explain the adverse effects associated with exhaust levels.
Round 2
Reviewer 2 Report
The work is good. However, I think the work "Lu et al. Combined effects of ambient air pollution and home environmental factors on low birth weight. Chemosphere 2020, 240: 124836" may provide a good reference to mention the limitation that they did not consider other type of air pollution or indoor air pollution.
Author Response
The work is good. However, I think the work "Lu et al. Combined effects of ambient air pollution and home environmental factors on low birth weight. Chemosphere 2020, 240: 124836" may provide a good reference to mention the limitation that they did not consider other type of air pollution or indoor air pollution.
ANSWER: We have added a sentence explaining that we are unable to adjust for indoor environmenton on lines 327 – 330, where we cite the paper by Lu et al.